

# Improved source apportionment of organic aerosols in complex urban air pollution using the multilinear engine (ME-2)

Qiao Zhu[1], Xiao-Feng Huang[1,*], Li-Ming Cao[1], Lin-Tong Wei[1], Bin Zhang[1], Lin-Yan He[1], Miriam Elser[2], Francesco Canonaco[2], Jay G. Slowik[2], Carlo Bozzetti[2], Imad El-Haddad[2], and André S.H. Prévôt[2]

[1]Key Laboratory for Urban Habitat Environmental Science and Technology, School of Environment and Energy, Peking University Shenzhen Graduate School, Shenzhen, 518055, China.

[2]Paul Scherrer Institute (PSI), 5232 Villigen-PSI, Switzerland

**Abstract** Organic aerosols (OAs), which consist of thousands of complex compounds emitted from various sources, constitute one of the major components of fine particulate matter. The traditional positive matrix factorization (PMF) method often apportions aerosol mass spectrometer (AMS) organic datasets into less meaningful or mixed factors, especially in complex urban cases. In this study, an improved source apportionment method using a bilinear model of the multilinear engine (ME-2) was applied to OAs collected during the heavily polluted season from two Chinese megacities located in the north and south with an Aerodyne high-resolution aerosol mass spectrometer (HR-ToF-AMS). We applied a rather novel procedure for utilization of prior information and selecting optimal solutions. Ultimately, six reasonable factors were clearly resolved and quantified for both sites by constraining one or more factors: hydrocarbon-like OA (HOA), cooking-related OA (COA), biomass burning OA (BBOA), coal combustion (CCOA), less-oxidized oxygenated OA (LO-OOA) and more-oxidized oxygenated OA (MO-OOA). In comparison, the traditional PMF method could not effectively resolve the appropriate factors, e.g., BBOA and CCOA, in the solutions. Moreover, coal combustion and traffic emissions were determined to be primarily responsible for the concentrations of PAHs and BC, respectively, through the regression analyses of the ME-2 results.

## 1 Introduction

Atmospheric aerosols are generating increasing interest due to their adverse effects on human health, visibility and the climate (IPCC, 2013; Pope and Dockery, 2006). Among different particulate compositions, many studies focus on organic aerosols (OAs) because they contribute 20-90% to the total submicron mass (Jimenez et al., 2009; Zhang et al., 2007). OAs can be either directly emitted by various sources, including anthropogenic (i.e., traffic and combustion activities) and biogenic sources, or produced via secondary formation after the oxidation of volatile organic compounds (VOCs) (Hallquist

---

[1] *Correspondence to*: X.-F. Huang (huangxf@pku.edu.cn)





et al., 2009). Therefore, the reliable source identification and quantification of OAs are essential before developing effective
political abatement strategies.
Aerodyne aerosol mass spectrometer (AMS) systems are the most widely adopted on-line aerosol measurement systems
for acquiring aerosol chemical compositions (Canagaratna et al., 2007; Pratt and Prather, 2012). An AMS provides on-line
quantitative mass spectra of non-refractory components from the submicron aerosol fraction with a high temporal resolution
(i.e., seconds to minutes) (Canagaratna et al., 2007). The total mass spectra can be assigned to both several inorganic
compounds and the organic fraction through mass spectral fragmentation tables (Allan et al., 2004). To further investigate the
different types of organic fractions, numerous studies have exploited the positive matrix factorization (PMF) algorithm and
apportioned the AMS organic mass spectra in terms of their source emissions or formation processes (Zhang et al., 2011).
PMF is a standard multivariate factor analysis tool (Paatero, 1999; Paatero and Tapper, 1994) that models the time series of
measured organic mass spectra as a linear combination of positive factor profiles and their respective time series. Most of the
earlier PMF studies were conducted on unit-mass resolution (UMR) mass spectrometers (Lanz et al., 2007; Lanz et al., 2010;
Ulbrich et al., 2009), although more have recently focused on high-resolution (HR) mass spectra PMF (Aiken et al., 2009;
Docherty et al., 2008; Huang et al., 2010). The use of HR mass spectra data to constrain PMF solutions can reduce their
rotational ambiguity and result in more interpretable OA factors. For example, Aiken et al. (2009) found that hydrocarbon-
like OA (HOA) and biomass burning OA (BBOA) were better separated using HR-AMS data than with UMR data. However,
even HR-AMS-PMF can also yield mixed factors (especially in heavily polluted areas) due to their complex emission
patterns.
The abundant characteristic fragments for cooking-related OA (COA) (e.g., $m/z$ 55 and 57) and coal combustion OA
(CCOA) (e.g., $m/z$ 51, 53, and 65) can be observed in the mass spectrum of the HOA factor (He et al., 2010; Hu et al., 2013).
Elser (et al., 2016) analyzed two urban HR-AMS datasets in China, and their PMF results showed an HOA profile that
contained a high concentration of $C_2H_4O_2^+$ ($m/z$ 60), which is a BBOA tracer ion. In addition, $CO_2^+$ ($m/z$ 44) contributed
more to COA compared to oxygenated OA (OOA). To solve this "mixed factor" problem in PMF analysis, some researchers
attempted to use the multilinear engine algorithm (ME-2) with user-provided constraints (Canonaco et al., 2013; Crippa et al.,
2014; Elser et al., 2016; Reyes-Villegas et al., 2016). However, several key issues with the ME-2 in these studies, such as
reliability of the user-input constraints and the criteria used to determine an optimal result, still require further investigation.
Most ME-2 studies (Crippa et al., 2014; Elser et al., 2016; Reyes-Villegas et al., 2016) were based on HR-AMS datasets and
utilized mass profiles of PMF results from Paris (mostly due to the lack of other reliable source profiles) and did not consider
the specific sampling sites, which could result in uncertainties.
In this study, a novel source apportionment technique using the multi-linear engine tool (ME-2) was successfully
applied to organic mass spectra obtained with an HR-ToF-AMS at two urban sites during pollution-heavy periods during the
same year. The improved OA source apportionment results are discussed and compared with an unconstrained PMF analysis.



## 2 Materials and methods

### 2.1 Sampling sites and period

Measurements at Qingdao (36.10 °N, 120.47 °E, 10 m above ground level, a.g.l.) were performed from 1 to 31 November 2013, while those in Dongguan were conducted from 12 December 2013 to 1 January 2014 (33.03 °N, 113.75 °E, 100 m a.g.l.). Qingdao is a coastal city with over 9 million inhabitants in northern China, while Dongguan has over 8 million inhabitants and is located in southern China (shown in Figure 1). Both of the sampling sites are on the tops of buildings in urban central areas. The surroundings include some agricultural counties, and thus, the sites are influenced by not only local urban emissions but also biomass burning from nearby farmlands.

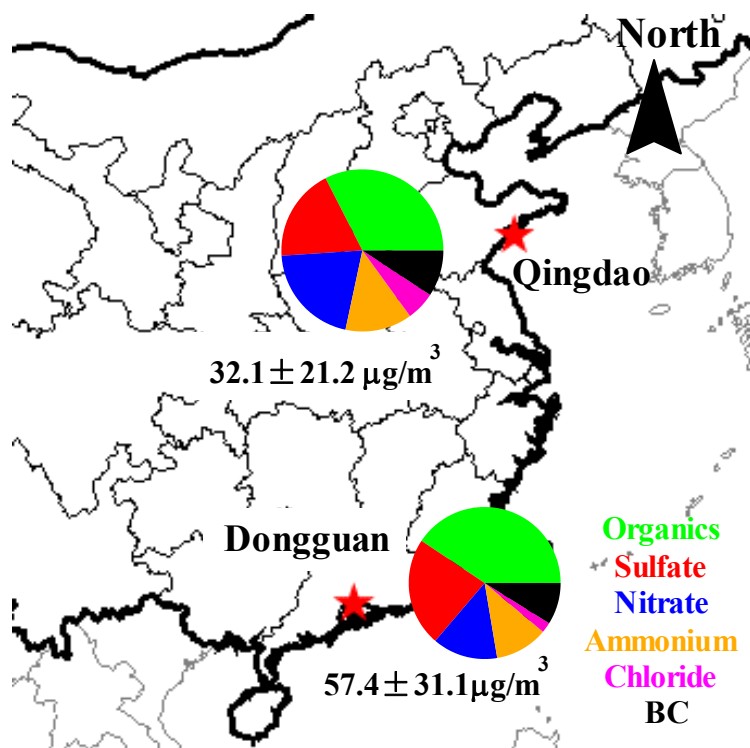

**Figure 1.** The locations and the average $PM_1$ chemical compositions of the Qingdao and Dongguan sampling sites.

### 2.2 Instrumentation

An HR-ToF-AMS was deployed for the on-line measurement of non-refractory $PM_1$ (Canagaratna et al., 2007). The setup and operation of the HR-ToF-AMS was similar to that in our previous studies (Huang et al., 2015; Huang et al., 2010). A $PM_{2.5}$ cyclone inlet was briefly placed on the roof of a building to remove coarse particles and to introduce an air stream containing the remaining particles into a room through a copper tube with a flow rate of 10 l min$^{-1}$. A nafion dryer (MD-070-



12S-4, Perma Pure Inc.) was positioned upstream of the HR–ToF–AMS to eliminate the potential influence of relative
humidity on the particle collection (Matthew et al., 2008), after which the HR–ToF–AMS isokinetically sampled from the
center of the copper tube at a flow rate of 80 ml min$^{-1}$. The instrument was operated at two ion optical modes with a cycle of
4 min, including 2 min for the mass-sensitive V-mode and 2 min for the high mass resolution W-mode. An aethalometer
(AE-31, Magee), which also has a PM$_{2.5}$ inlet, was simultaneously used for measurements of refractory black carbon (BC)
with a temporal resolution of 5 min.
A routine analysis of the HR–ToF–AMS data was performed using the software SQUIRREL (version 1.57) and PIKA
(version 1.16) written in Igor Pro 6.37 (Wave Metrics
Inc.)(http://cires1.colorado.edu/jimenezgroup/ToFAMSResources/ToFSoftware_/index.html). The ionization efficiency (IE)
was calibrated using pure ammonium nitrate particles following standard protocols (Drewnick et al., 2005; Jayne et al.,
2000). The relative IEs (RIEs) for organics, nitrate and chloride were assumed to be 1.4, 1.1 and 1.3, respectively. A
composition-dependent collection efficiency (CE) was applied to the data based on the method of Middlebrook et al. (2012)
and an organic elemental analysis was performed using the latest approach recommended by Canagaratna et al. (2015). The
mass concentrations of polycyclic aromatic hydrocarbons (PAHs) were quantitatively determined from the HR-AMS data
using the method of Bruns et al. (2015).

## 2.3 PMF and ME-2 methods for OA source apportionment

PMF is a mathematical technique used to solve bilinear unmixing problems (Paatero and Tapper, 1994) that enables a
description of the variability of a multivariate database as the linear combination of static factor profiles and their
corresponding time series. The bilinear factor analytic model in matrix notation is defined in Eq. (1), where the measured
matrix X (consisting of i rows and j columns) is approximated by the product of G (containing the factor time series) and F
(the factor profiles). E denotes the model residuals. The entries in G and F are fitted using a least-squares algorithm that
iteratively minimizes the quantity Q (Eq. 2), which is defined as the sum of the squared residuals ($e_{ij}$) weighted by their
respective uncertainties ($\sigma_{ij}$).

$$X = G \times F + E \qquad (1)$$

$$Q = \sum_{i=1}^{m} \sum_{j=1}^{n} \left( \frac{e_{ij}}{\sigma_{ij}} \right)^{2} \qquad (2)$$

In this study, we adopted SoFi (Canonaco et al., 2013), which is an implementation of the multilinear engine (ME-2)
(Paatero, 1999), to perform the organic HR-AMS data analysis. In contrast to an unconstrained PMF analysis, ME-2 enables
a more complete exploration of the rotational ambiguity of the solution space. In our case, this is achieved by directing the
solution towards environmentally meaningful rotations using the $a$ value approach. This method uses prior input profiles and
the scalar a to constrain one or more output factor profiles such that they fall within a predetermined range. The $a$ value
determines the extent to which the output profiles are allowed to vary from the input profiles according to Eq. (3), where f
represents the factor profile and j indicates the $m/z$ of the ions.



$$f_{j,solution} = f_j \pm a \times f_j \qquad (3)$$

The number of output factors, which is selected by the user, is a key consideration for PMF analysis. Most unconstrained
PMF results were chosen following the procedures detailed in Zhang et al. (2007). However, additional outputs in ME-2 can
be generated to explore more of the solution space, and more criteria should be developed to support the factor identification,
which will be discussed in section 3.
**3 Results and discussion**
In this section, a conventional PMF without any prior information is performed to analyze the OA sources. Then, we
use the ME-2 method to optimize the OA source apportionment based on the information obtained from the PMF method.
Finally, the improved source apportionment results derived using ME-2 are further discussed and analyzed.
**3.1 OA source apportionment using an unconstrained PMF method**
We performed unconstrained runs with a range from two to ten factors. Generally, PMF solutions with large numbers of
factors are not considered due to possible mathematical splits of the factor profiles. However, some factors that have small
contributions or that have similar mass profiles as other factors (but different time series) may only be found in solutions
with large numbers of factors. We observe that most of the solutions provided via PMF include either multiply split factors
or mixed factors that are not properly separated from one another. In other words, PMF does not produce an appropriate
solution. The 6-factor solutions for Qingdao and Dongguan are shown in Figure S1 and S2, and three types of primary OAs
(POAs) were identified for each sampling site, including HOA), coal combustion OA (CCOA) and cooking OA (COA) for
Qingdao and HOA, biomass burning OA (BBOA) and COA for Dongguan. Oxygenated OA (OOA) seems to be excessively
split in the 6-factor solutions for both of the sites. HOA is distinguished by alkyl fragment signatures with prominent
contributions of $m/z$ 55 ($C_4H_7^+$) and $m/z$ 57 ($C_4H_9^+$) (Ng et al., 2011). The COA profile is similar to that of HOA but has a
higher contribution from oxygenated ions at $m/z$ 55 ($C_3H_3O^+$) and $m/z$ 57 ($C_3H_5O^+$) (Mohr et al., 2012). BBOA is
characterized by the presence of signals at $m/z$ 60 ($C_2H_4O_2^+$) and $m/z$ 73 ($C_3H_5O_2^+$), which are identified as fragments from
anhydrous sugars present in biomass smoke (Alfarra et al., 2007). The OOA profile is characterized by a high signal at m/z
($CO_2^+$). Note that some POA profiles in this solution indicate mixing; for example, CCOAs in Qingdao contain a high
concentration of the biomass burning tracer ion ($m/z$ 60, $C_2H_4O_2^+$), and HOAs in Dongguan have a higher-than-expected
contribution of $m/z$ 44 ($CO_2^+$) with a high O/C ratio (0.26). In addition, CCOA seems to be mixed with BBOA. We then
further verified the solutions with additional factors. The results show that BBOA and CCOA are separated from each other
in the 7- and 8-factor solutions for Qingdao (see Figure S1) and that better signals for unmixed and stable HOA with low
O/C ratios of 0.17 or 0.18 emerged in the 7- to 10-factor solutions for Dongguan (see Figure S2).
**3.2 Improved OA source apportionment using the ME-2 method**
Before operating ME-2, feasible and reasonable prior input profiles must be determined. To the best of our knowledge,
this is the first HR-OA data set that employs anchor profiles extracted from an unconstrained PMF solution with a higher





number of factors, and the same approach has been successfully applied to source apportionment efforts using UMR ME-2
(Fröhlich et al., 2015). In our case for Qingdao, the BBOA factors from the 7- and 8-factor solutions may be used as anchor
profiles. Although these two BBOA factors are quite similar, the BBOA from the 8-factor solution is better suited to be a
constraining profile due to its smaller $m/z$ 44 ($CO_2^+$) signal and higher $m/z$ 60 ($C_2H_4O_2^+$) signal (see Figure S3). In addition,
the BBOA from the 8-factor solution also correlates better with the BBOA from a Chinese biomass burning simulation
($R^2$=0.81) than the 7-factor solution ($R^2$=0.79) (He et al., 2010). Moreover, the best interpretable results from the data set are
the 6-factor solutions with factors that include HOA, COA, BBOA, CCOA, less-oxidized oxygenated OA (LO-OOA) and
more-oxidized oxygenated OA (MO-OOA) (Figure 3a). Considering $a$ values between 0 and 1 with a step of 0.1 for BBOA
yields 11 possible solutions, and some criteria were established to obtain a better environmental OA source apportionment.
In this study, we used two simple and reasonable criteria: the reasonability of the O/C ratio and the correlation between the
factors and the tracers. The O/C ratios for six resolved factors and the correlations between CCOA and PAHs, HOA and BC
for 11 solutions with different $a$ values are shown in Table S2. These results indicate that all of the O/C ratios for each factor
and each factor-tracer correlation are quite similar to one another and that they agree with the range of values in the literature
(Canagaratna et al., 2015). Therefore, the solutions averaged over the 11 outputs were considered the final results for
Qingdao.
The anchor profile for HOA for Dongguan can be obtained from unconstrained PMF solutions. The averaged HOA
profile from the 7- to 10-factor solutions was used as the anchor profile for ME-2 due to the small differences among the
different solutions. Additionally, the constraining CCOA profile for Dongguan is still under consideration because the mass
spectrum of BBOA was found to be very similar to that of CCOA, raising the concern that coal combustion particles might
have been incorrectly apportioned to biomass burning sources (Wang et al., 2013). Furthermore, an appropriate CCOA
anchor profile could not be obtained due to an increase in the unconstrained PMF factor number (see Figure S2) and because
few studies have reported a comparison of the CCOA profile to other PMF factors. The best approach is to employ the
CCOA profile from Qingdao as the constraining profile for Dongguan in ME-2, as these two campaigns were conducted
using the same HR-ToF-AMS in the same year and because coal combustion is a significant source of OAs in Qingdao due
to domestic heating during the wintertime. In addition, the CCOA from Qingdao has a very good correlation ($R^2$=0.97) with
CCOA profiles reported at other Chinese urban sites (Elser et al., 2016) (see Figure S4). The input profiles for HOA and
CCOA in Dongguan prior to operating ME-2 are shown in Figure 2. The $a$ values were set from 0 to 1 with an increment of
0.1 for both HOA and CCOA. All of the O/C ratios for HOA, CCOA, COA and BBOA among the 121 possible solutions are
listed in Table S3. The O/C ratio of HOA in the unconstrained PMF results remained between approximately 0.17 and 0.18,
providing a filter criterion with which to assess reasonable ME-2 solutions, and only solutions with $a$ values between 0 and
0.1 fell into this range (Table S1). The O/C ratios of other factors for $a$ values between 0 and 0.1 are shown in Table S3. The
solutions using $a$ values between 0 and 0.1 for the HOA profile and an $a$ value of 0.9 for the CCOA profile are considered
ideal results for three reasons. First, unlike the HOA mass spectra, CCOAs from different sites show higher variability and
the CCOA anchor profile is not derived from itself, and therefore, it is reasonable to restrict the constraint with small $a$





values for HOA and a looser constraint should be applied for CCOA, which is consistent with the *a* values selecting rules in
London ME-2 study (Reyes-Villegas et al., 2016). Second, the POA factors in Dongguan, including HOA and CCOA, have
higher O/C ratios likely as a result of a higher atmospheric oxidizing capacity and a stronger photochemical formation in
Southern China (Hofzumahaus et al., 2009). Moreover, some studies reported that BBOAs undergo substantial chemical
processing immediately after emission and that aged BBOAs had significant concentrations in fresh plumes (Zhou et al.,
2017). Thus, CCOAs in Dongguan are very likely to demonstrate relatively higher ages than those in Qingdao (0.15) with
higher O/C ratios (but with an O/C ratio of up to 1.25 when the *a* value is 1, which is unacceptable). Third, with an increase
in the *a* value for CCOA, two types of OOAs become more distinctive, and the factor correlates better with the tracer (Table
S1 and Table S4).

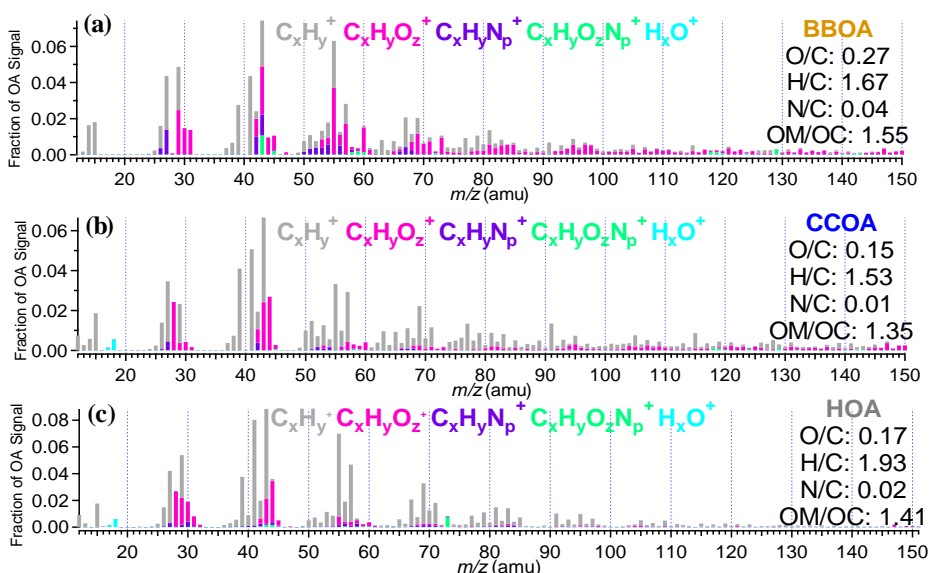

**Figure 2.** The anchor mass spectra for (a) BBOA, (b) CCOA and (c) HOA in the ME-2 analysis.
**3.3 Variations in the OA factors**
Figure 1 shows the chemical compounds of $PM_1$, including the non-refractory (NR) components measured via HR-
AMS (i.e., OA, $SO_4$, $NO_3$, $NH_4$ and Cl) and BC concentrations measured via the AE-31, during the sampling period in both
Qingdao and Dongguan. The average $PM_1$ mass concentration was $32.1 \pm 21.2$ µg/m$^3$ (mean ± standard deviation) in Qingdao
and $57.4 \pm 31.1$ µg/m$^3$ in Dongguan. The temporal variations in the $PM_1$ species in conjunction with meteorological
parameters are shown in Figure S5. Although Dongguan is located in southern China with relatively less air pollution
(Huang et al., 2012), the $PM_1$ mass concentration was higher. This is mainly because of stagnant meteorological conditions
with low average wind speeds (i.e., 2.3 m/s) and a maximum wind speed of less than 6 m/s. Among the $PM_1$ compounds,





OAs accounted for 32.5% of $PM_1$ in Qingdao and 40.6% in Dongguan. This suggests that OA constitutes a very important
fraction at both urban sites. Thus, the final and detailed results of the OA source apportionment are presented in this section.
For Qingdao, the final result is the average of all of the ME-2 runs with constraints including *a* values from 0 to 1 fulfilling
the criteria described in section 3.2. The mass spectra and time series of the resolved OA sources are shown in Figure 3a.
The characteristics of each factor were distinct. The BBOA profile contained the highest *m/z* 60 fraction *f*60 (1.5%)
compared to the other factors, and the concentrations were highly correlated with $C_2H_4O_2^+$ ($R^2$=0.81). The mass spectra of
COA was characterized by a high *m/z* 55/57 ratio, which is consistent with previous results (He et al., 2010; Mohr et al.,
2012; Sun et al., 2016). In addition, the time series of COA showed a good correlation with its tracer ion $C_6H_{10}O^+$ in
accordance with (Sun et al., 2016). HOAs were correlated well with BC ($R^2$=0.65), and CCOAs were highly correlated with
PAHs ($R^2$=0.94). Among the two types of OOAs, the less-oxidized OOA (LO-OOA) had a lower $CO_2^+$ fraction and O/C
ratio (0.62) compared with the more oxidized OOA (MO-OOA), which had a higher $CO_2^+$ fraction and O/C (0.91) ratio. The
sum of LO-OOA and MO-OOA showed a high correlation with the sum of sulfate and nitrate ($R^2$=0.76). The POAs
(including HOA, COA, BBOA and CCOA) contributed 53.4% to the OA concentration (Figure 3a), which was almost equal
to the SOA fraction. In terms of the diurnal trends of the OA factors shown in Figure 3a, they are all partially driven both by
PBL dynamics (demonstrating an increased dilution during the daytime and an accumulation of particulate matter overnight)
and by the diurnal emission profile. The diurnal trend of HOA showed pronounced peaks during the morning and evening
rush hours (8:00-9:00 and 19:00-21:00), which is typically the case for traffic-related pollutants. COA shows a very distinct
daily trend with strong peaks during the lunch (approximately 12:00) and dinner (19:00-20:00) periods. CCOAs constituted
an important and dominant source of pollutants during the wintertime in northern Chinese areas (Elser et al., 2016) due to
heating activities, especially with regard to the central-heating supply that began on November 13 and continued until the
end of the campaign. The diurnal variations of the four POA factors before and during the central-heating period are shown
in Figure S6. In comparison with the other three POAs, the diurnal pattern of CCOA showed a clear increase during the
central-heating period with concentration peaks during the morning (at approximately 9:00) and at night (starting to rise at
18:00), which seems consistent with heating emissions and atmospheric dilution. The diurnal trends of BBOA were similar
to those of CCOA. The dilution of these particles within a deeper PBL during the daytime resulted in a decreasing trend in
the BBOA concentration, while peaks related to residential heating were observed during the morning (between 09:00 to
10:00) and at night (starting to rise at 17:00). The main difference between the LO-OOA and MO-OOA diurnal patterns is
that an increase in the MO-OOA mass concentration was observed during the daytime, implying that the formation of
secondary organic aerosols was greatly enhanced during the afternoon. In addition, the diurnal cycle for LO-OOA showed a
relatively smaller decrease during the daytime compared with the POA factors. These characteristics of the OOA diurnal
trend confirm their secondary nature.

For Dongguan, similar to the OA source apportionment using ME-2 in Qingdao, the final result is the average of two

accepted a-value solutions with six identified factors, including HOA, CCOA, COA, BBOA, LO-OOA and MO-OOA. All of
the information regarding the final source results is shown in Figure 3b. Good correlations between each OA factor and their





tracers indicate that the resolved ME-2 results are reasonable. Note that a few sharp drops (which always occurred at
approximately 20:00) were observed in the MO-OOA time series ranging from December 29 to January 5, which coincides
with extreme organic aerosol pollution (Figure S5). The inherent mechanisms for these drops remain unexplained, although
we have tried a number of reasonable approaches (e.g., splitting the period into sub-periods to identify the sources,
constraining more factors before running ME-2, and examining more factors) to address this issue. A similar problem in the
MO-OOA time series was also found in a recent ME-2 application (Qin et al., 2017). In our case, we presume this might be
the result of relatively worse meteorological conditions at night during the sampling period, thereby increasing the
contribution of late supper emissions and leading to the overestimation of COAs offset by drops in the MO-OOA
concentration. Also note that the O/C ratios of the POAs in Dongguan were higher than those in Qingdao, suggesting that
POA emissions in Dongguan underwent faster chemical processing. In addition, the relatively smaller contributions of POAs
further support this inference. Freshly emitted POAs may get mixed with aged OAs more easily, while ME-2 may still
consider them unmixed. MO-OOAs accounted for an average of 42.8% of the total OA mass (which is much greater than the
contribution of LO-OOAs), which is probably because some POA species could have been rapidly converted to very aged
OOAs (Bougiatioti et al., 2014; Xu et al., 2015). As mentioned above, the characteristics of the diurnal trends of the POA
factors in Dongguan were similar to those in Qingdao, and thus, we focused on the OOA factors. MO-OOAs still showed
higher concentrations during the daytime but, unlike LO-OOAs in Qingdao, the diurnal patterns of LO-OOAs in Dongguan
were flat, implying that secondary OA formation in the LO-OOAs basically offset the influences of PBL variations.
Meteorological conditions (especially wind) play a crucial role in the dilution and transport of air pollution. We used
the relationships between the component concentrations and wind to profoundly understand the origins of the OA factors and
their nature. The distributions of the OA factor concentrations versus the wind direction and speed are plotted in Figure S7.
For both of the urban sites, higher mass concentrations of the POA factors were mostly accompanied by low wind speeds,
denoting their local emission characteristics. Additionally, for the OOA factors, a large proportion of their higher
concentrations were maintained at higher wind speeds, indicating that the OOAs were formed by transport processes.
However, the small fraction of high-level OOAs that was concentrated within the low wind-speed region represents the fast
formation of OOAs from some local POA.


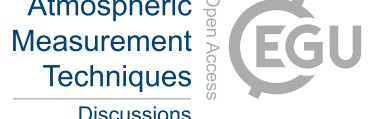

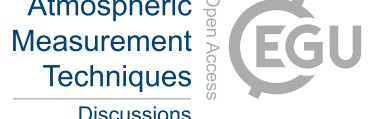

**Figure 3.** Mass spectra of the OA factors, average fractions of the OA factors, diurnal variations of the OA factors and time series of the OA factors identified by the ME-2 method for (a)Qingdao and (b) Dongguan.






### 3.4 Regression analysis for POA tracers

BC and PAHs are mainly derived from incomplete combustion processes (Schmidt and Noack, 2000; White, 1985), and thus, they were used as tracers for the POAs. In this study, the BC was directly measured by the AE-31, and the PAHs were quantified using the method developed by Bruns et al. (2015) based on AMS data. Both the BC and PAHs showed pronounced diurnal cycles similar to those of the POAs (see Figure S8). In addition, POAs are properly split into different subtypes via the ME-2 method, thereby providing the possibility to better understand the contributions of different POAs to BC and PAHs and to verify the POA source identification. In this section, we use a multi-linear regression method to analyze the POA factors for BC and PAHs.

Figure 4 shows the average contributions of OA sources to BC and PAHs in Qingdao and Dongguan. At both sites, HOAs were the dominant attribute of BC (51% for Qingdao and 40% for Dongguan) and CCOAs contributed the most to the PAHs (59% for Qingdao and 43% for Dongguan), indicating that BC mainly originates from traffic emissions and that PAHs in the Chinese urban polluted atmosphere are dominated by coal combustion during the wintertime. These findings are consistent with results reported in similar studies (Elser et al., 2016; Huang et al., 2015; Huang et al., 2010; Sun et al., 2016; Xu et al., 2014; Zhang et al., 2008). Moreover, the ratio of PAHs to OAs (1.8%) in Qingdao was similar to that in the northern Chinese urban site of Xi'an (1.9%) (Elser et al., 2016) but was higher than that in Dongguan (0.9%). This is likely because a larger fraction of coal combustion to the total OA concentration would enhance the ratio of PAHs to OAs (Elser et al., 2016). Biomass burning was the second-most important source for both BC and PAHs; it was responsible for 33% and 29% of the BC at Qingdao and Dongguan, respectively, and for 29% and 34% of the PAHs at Qingdao and Dongguan, respectively. Cooking emissions were a minor source of BC and PAHs, accounting for less than 10%. These results also correspond with published findings. For example, biomass burning is an important source for BC (Kondo et al., 2011; Reddy et al., 2002) and, in some regions with fewer traffic emissions, BC has the best correlation with BBOAs (Schwarz et al., 2008). In addition, in Beijing and California, PAHs are correlated well with BBOAs but are much more weakly correlated with COAs (Ge et al., 2012; Hu et al., 2016; Sun et al., 2016).



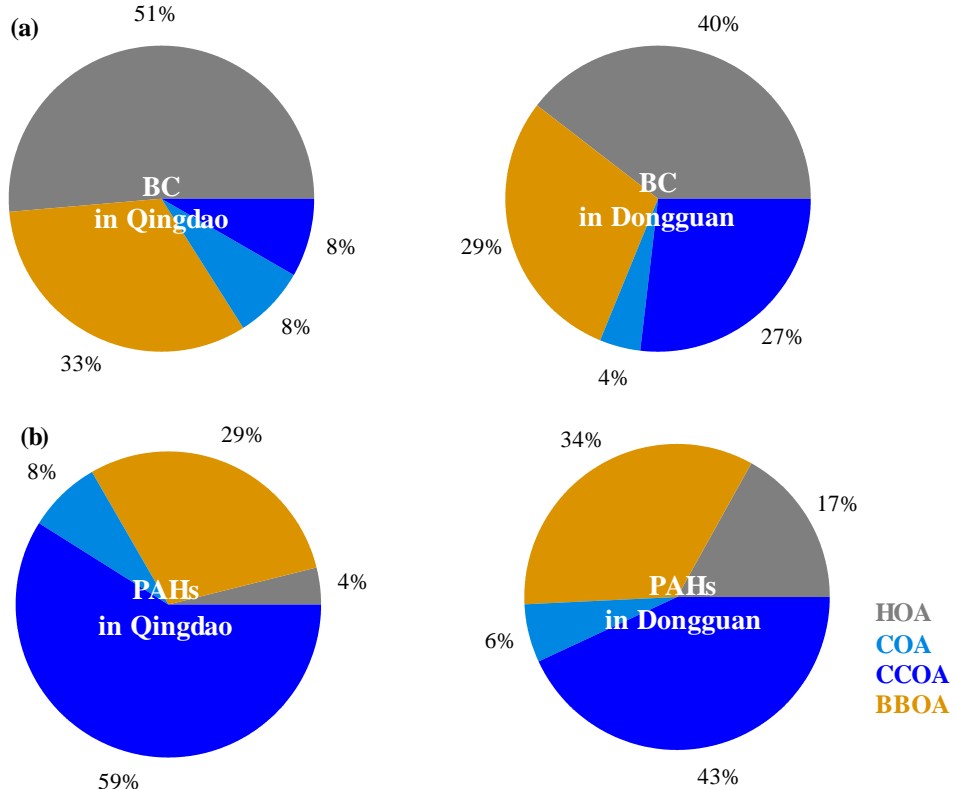


**Figure 4.** (a) Average contributions of POA factors to BC; (b) average contributions of POA factors to PAHs.
**4 Conclusions**
In this study, we used PMF to interpret the pollutants of two heavily polluted urban cities, and we found that PMF does
not work properly (i.e., it does not allow for the separation of several primary sources of OAs). Therefore, we adopted the
ME-2 approach, which yields more reliable solutions. Technically, there are three important steps when using the ME-2
method to interpret the sources of OAs. The first step is to investigate the mixed and unidentified factors that are constrained
according to issues in the unconstrained PMF results. Generally, we constrained one or more POA factors (i.e., HOA, COA,
BBOA and CCOA) for the polluted urban sites. The second step is to search for a reasonable anchor profile for each
constrained factor. Two approaches were used: searching for anchor profiles via an increase in the number of unconstrained
PMF factors from the same data set and using mass profiles derived from other similar studies. The third step is to choose
the criteria for obtaining the optimal results. The choice of a reasonable range of O/C ratios may represent a good criterion
for HR-OA apportionment since the O/C ratio is a significant and distinctive characteristic for different OA factors. In
addition, correlations between the resolved OA factors and their tracers were also suggested.



## Acknowledgments

This work was supported by the National Natural Science Foundation of China (U1301234, 41622304), the Ministry of Science and Technology of China (2017YFC0210004), and the Science and Technology Plan of Shenzhen Municipality.

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
