# Peer review of "I Improved source apportionment of organic aerosols in complex"

_Atmospheric Measurement Techniques, 2017_

## Referee Comment (RC1) · Anonymous Referee #2 · 23 Nov 2017

This manuscript presents a source apportionment study on the organic aerosols (OA) collected by an Aerosol Mass Spectrometer in two urban environments in China. The study compares the source profiles and temporal variations of factors (sources) derived from a "traditional" Positive Matrix Factorization (PMF) method versus a more advanced Multilinear Engine (ME-2) method with constrained parameters. The authors found that the ME-2 method yielded more reliable results in terms of separation of different primary OA. Six OA factors were resolved using the ME-2 techniques. My general concern about the PMF technique on AMS OA spectra is that, is it reasonable to assume the number of factors and their profiles are always the same in different locations? Is it mathematically true that the more factors you allow PMF to resolve, the

better the overall results would be? For example, in Line 133-135, the authors stated that BBOA and CCOA factors could be properly resolved by traditional PMF when the total number of factors was flexed up to 10. Also, the BBOA in Dongguan seems to be much more oxygenated than that in Qingdao, suggested by the much higher O/C ratio and OM/OC ratios. Could that be due to some factor mixing (i.e. BBOA mixed with OOA), or is it more reasonable to not assume they are both BBOA? E.g., maybe the Dongguan-BBOA should be characterized as aged BBOA or something similar? What are the reported ranges of O/C and OM/OC ratios for BBOA in literature?

Other comments: Section 2.1: Are there any industrial sources near the sampling sites at both cities? If so, please specify.

Figure 3: I highly recommend the authors reorder the source profiles and time series plots in (a) and (b) so that they follow the same order for both sites for easier cross-comparison.

Line 259: Please add some more details on how PAHs are derived from the OA spectra. You can also consider moving the details to Section 2.

Line 269: Are the ratios of PAHs to COA at the two sites in this study comparable to each other and other cities in the literature? Is there any compositional difference in the coals used for heating (and/or cooking) in the e.g. Northern and Southern China?

Line 275: correspond with -> are consistent with

Line 282: pollutants -> organic aerosol pollutants
* * *

---

## Referee Comment (RC2) · Anonymous Referee #1 · 29 Nov 2017

This paper details the use of a combination of unconstrained PMF runs and ME-2 to provide better factorisation of AMS/ACSM data without the need to rely on a priori spectra as much. The basics are to perform and unconstrained PMF run, select a subset of the factors that would appear to be stable and then use these as target factors for ME-2 such that the solver can be run with fewer degrees of freedom and thus give a less ambiguous output. While I could envisage that the success of this technique may vary dataset to dataset, it may be that the technique could be further refined and best practice established in due course, so I would consider this worthy as a technical paper.

The manuscript is generally suitable for AMT and very well written and I can see the hypothetical benefit to the technique. However, I find it a little curious to see that one of the factors identified for this treatment is BBOA, which is known to vary within individual datasets (e.g. Young et al. (2015) Atmos. Chem. Phys., 15: 2429-2443, 10.5194/acp-15-2429-2015). It's also curious that they should select the optimum based on comparisons of the mass spectrum with previous studies. By doing this, I would see that what they are doing is little different to simply using the a priori reference spectrum in ME-2, thus defeating the whole purpose of the technique. The authors should comment on this.

As a more general point, I would request a few changes to be made to give the paper a better theoretical footing and also to be of more use to potential readers, which are necessary for this to be a good technical paper. It may be that this technique is more suitable for some datasets than others, so it is important to do more than simply show it working in one instance. I would recommend that the authors do the following:

1) Present a more robust theoretical case for the improvement in apportionment that could result from this method. While I would not ask the authors to submit a full mathematical proof, I would surmise that the factors that this would work best for this treatment would be the ones whose profiles are invariant (i.e. conform to the PMF data model) and produce a time series that is distinct from the other components. These should be explicitly stated and the implications of using factors that do not conform to these assumptions discussed. For instance, I would expect that if a factor has a profile that varies with time, one would expect that this would be under-represented in the unconstrained PMF solution (with some of its variability being represented by other factors) and therefore under-represented in the ME-2 solution.

2) A step-by-step recommended procedure should be unambiguously presented, for the benefit of those attempting to recreate the method. While this is kind-of done in the conclusions, it is very vague in places.

[Figure]

As a final technical query, can the authors confirm that the PAH data used to validate the result were not allowed to influence the factorisation originally? It would defeat the object of the exercise if they were.

---

## Author Comment (AC1) · 24 Dec 2017

1. My general concern about the PMF technique on AMS OA spectra is that, is it reasonable to assume the number of factors and their profiles are always the same in different locations? Is it mathematically true that the more factors you allow PMF to resolve, the better the overall results would be? For example, in Line 133-135, the authors stated that BBOA and CCOA factors could be properly resolved by traditional PMF when the total number of factors was flexed up to 10. Also, the BBOA in Dongguan seems to be much more oxygenated than that in Qingdao, suggested by the much higher O/C ratio and OM/OC ratios. Could that be due to some factor mixing (i.e. BBOA

mixed with OOA), or is it more reasonable to not assume they are both BBOA? E.g., maybe the Dongguan-BBOA should be characterized as aged BBOA or something similar? What are the reported ranges of O/C and OM/OC ratios for BBOA in literature?

REPLY: The numbers and types of factors were determined according to the unconstrained PMF results for each case, and could vary for different cases. For both the two sites in this paper, a 6-factor solution is chosen as the final result following the procedures detailed in Zhang et al. (2007). Crippa et al. (2014) also provides some guidelines to identify HOA, COA (check f55/f57) and BBOA (f60) for ME-2, and the detailed information to identify the existence of CCOA is presented in our paper. Therefore, we have enough evidence to expect four POA factors and two OOA factors for both sites in this study, and finally we got satisfactory running results. More OA factors output by ME-2 would not produce more significant factors. On the other hand, the purpose that we allow PMF to resolve more numbers is to find "purer" or more reasonable MS profiles (e.g., with a reasonable O/C ratio) for certain factors that were not well identified previously, not to look for a better overall result of PMF. The range of O/C ratio and OM/OC ratio of BBOA reported in Canagaratna et al. (2015) is from 0.25 to 0.55, and from 1.50 to 1.88, respectively. But the fresh BBOA can be rapidly converted to OOA in less than 1 day (Bougiatioti et al., 2014), where the O/C ratio for aged BBOA could be up to 0.85 (Zheng et al., 2017). BBOA in Dongguan was apparently not fresh, considering it is an urban site and Dongguan has a warmer ambient air even in winter (17 °C in Dongguan; 9 °C in Qingdao), therefore the BBOA factor identified in Dongguan, with a strong contribution of m/z 60, has a higher O/C, indicating it is an aged and oxygenated BBOA. Following the suggestion of this reviewer, we will name this factor as Aged-BBOA in the revised paper.

2. Are there any industrial sources near the sampling sites at both cities? If so, please specify.

REPLY: There is no industrial sources around the sampling sites, which has been clarified in the text.

3. Figure 3: I highly recommend the authors reorder the source profiles and time series plots in (a) and (b) so that they follow the same order for both sites for easier cross comparison.

REPLY: We have corrected it.

4. Line 259: Please add some more details on how PAHs are derived from the OA spectra. You can also consider moving the details to Section 2.

REPLY: We have added the details about the process of PAH quantification in Section 2.4.

5. Line 269: Are the ratios of PAHs to COA at the two sites in this study comparable to each other and other cities in the literature? Is there any compositional difference in the coals used for heating (and/or cooking) in the e.g. Northern and Southern China?

REPLY: We didn't mention the ratios of PAHs to COA but the ratios of PAHs to OAs, and the ratio of PAHs to OAs (1.8%) in Qingdao was similar to that in the northern Chinese urban site of Xi'an (1.9%) (Elser et al., 2016) but was higher than that in Dongguan (0.9%) in Southern China. According to the spatial variation of heavy metal elements from coal (Tian et al., 2012), we can find that the emission compositions of coal combustion in different regions in China are quite similar. This information has been added into the revised text.

6. Line 275: correspond with -> are consistent with

REPLY: We have corrected it.

7. Line 282: pollutants -> organic aerosol pollutants

REPLY: We have corrected it.

References

Bougiatioti, A., I. Stavroulas, E. Kostenidou, P. Zarmpas, C. Theodosi, G. Kouvarakis, F.

Canonaco, A. S. H. Prévôt, A. Nenes, S. N. Pandis, and N. Mihalopoulos : Processing of biomass-burning aerosol in the eastern Mediterranean during summertime, Atmos. Chem. Phys., 14(9), 4793-4807, doi: 10.5194/acp-14-4793-2014,2014.

Canagaratna, M. R., J. L. Jimenez, J. H. Kroll, Q. Chen, S. H. Kessler, P. Massoli, L. Hildebrandt Ruiz, E. Fortner, L. R. Williams, K. R. Wilson, J. D. Surratt, N. M. Donahue, J. T. Jayne, and D. R. Worsnop : Elemental ratio measurements of organic compounds using aerosol mass spectrometry: characterization, improved calibration, and implications, Atmos. Chem. Phys., 15(1), 253-272, doi: 10.5194/acp-15-253-2015,2015.

Crippa, M., F. Canonaco, V. a. Lanz, M. Äijälä, J. D. Allan, S. Carbone, G. Capes, D. Ceburnis, M. Dall'Osto, D. A. Day, P. F. DeCarlo, M. Ehn, a. Eriksson, E. Freney, L. Hildebrandt Ruiz, R. Hillamo, J. L. Jimenez, H. Junninen, A. Kiendler-Scharr, A. M. Kortelainen, M. Kulmala, A. Laaksonen, A. A. Mensah, C. Mohr, E. Nemitz, C. O'Dowd, J. Ovadnevaite, S. N. Pandis, T. Petäjä, L. Poulain, S. Saarikoski, K. Sellegri, E. Swietlicki, P. Tiitta, D. R. Worsnop, U. Baltensperger, and A. S. H. Prévôt: Organic aerosol components derived from 25 AMS data sets across Europe using a consistent ME-2 based source apportionment approach, Atmos. Chem. Phys., 14(12),6159-6176, doi: 10.5194/acp-14-6159-2014, 2014.

Elser, M., R. J. Huang, R. Wolf, J. G. Slowik, Q. Wang, F. Canonaco, G. Li, C. Bozzetti, K. R. Daellenbach, Y. Huang, R. Zhang, Z. Li, J. Cao, U. Baltensperger, I. El-Haddad, and A. S. H. Prévôt : New insights into PM2.5 chemical composition and sources in two major cities in China during extreme haze events using aerosol mass spectrometry, Atmos. Chem. Phys., 16(5), 3207-3225, doi: 10.5194/acp-16-3207-2016,2016.

Tian, H., Cheng, K., Wang, Y., Zhao, D., Lu, L., Jia, W. and Hao, J.: Temporal and spatial variation characteristics of atmospheric emissions of Cd, Cr, and Pb from coal in China. Atmospheric Environment, 50: 157-163,doi: 10.1016/j.atmosenv.2011.12.045,2012.

Zhang, Q., J. L. Jimenez, M. R. Canagaratna, J. D. Allan, H. Coe, I. Ulbrich, M. R. Alfarra, A. Takami, A. M. Middlebrook, Y. L. Sun, K. Dzepina, E. Dunlea, K. Docherty, P. F. DeCarlo, D. Salcedo, T. Onasch, J. T. Jayne, T. Miyoshi, A. Shimono, S. Hatakeyama, N. Takegawa, Y. Kondo, J. Schneider, F. Drewnick, S. Borrmann, S. Weimer, K. Demerjian, P. Williams, K. Bower, R. Bahreini, L. Cottrell, R. J. Griffin, J. Rautiainen, J. Y. Sun, Y. M. Zhang, and D. R. Worsnop :Ubiquity and dominance of oxygenated species in organic aerosols in anthropogenically-influenced Northern Hemisphere midlatitudes, Geophys. Res. Lett., 34(13), L13801, doi: 10.1029/2007GL029979,2007.

Zheng, J., Hu, M., Du, Z., Shang, D., Gong, Z., Qin, Y., Fang, J., Gu, F., Li, M., Peng, J., Li, J., Zhang, Y., Huang, X., He, L., Wu, Y., and Guo, S.: Influence of biomass burning from South Asia at a high-altitude mountain receptor site in China, Atmos. Chem. Phys., 17, 6853-6864, https://doi.org/10.5194/acp-17-6853-2017, 2017.

---

## Author Comment (AC2) · 24 Dec 2017

1. However, I find it a little curious to see that one of the factors identified for this treatment is BBOA, which is known to vary within individual datasets (e.g. Young et al. (2015) Atmos. Chem. Phys., 15: 2429-2443, 10.5194/acp-15-2429-2015). It's also curious that they should select the optimum based on comparisons of the mass spectrum with previous studies. By doing this, I would see that what they are doing is little different to simply using the a priori reference spectrum in ME-2, thus defeating the whole purpose of the technique.

REPLY: Although BBOA varies across different datasets, the differences among different BBOAs are much less than those among different OA factors, which made BBOA identified by factorization in many studies. In this study, the purpose of comparing the BBOA anchor profiles from the unconstrained PMF results with the previous ones was just to confirm their basic BBOA characteristics, providing a new way to obtain a reasonable anchor profile for the ME-2 method, without the need to rely on a priori spectra. In the revised text, we have added the analysis that using the BBOA (and other POAs) spectra generated by the unconstrained PMF run of the same local dataset was indeed better than using a priori spectra from other studies, as in the reply to the next question.

2. Present a more robust theoretical case for the improvement in apportionment that could result from this method. While I would not ask the authors to submit a full mathematical proof, I would surmise that the factors that this would work best for this treatment would be the ones whose profiles are invariant (i.e. conform to the PMF data model) and produce a time series that is distinct from the other components. These should be explicitly stated and the implications of using factors that do not conform to these assumptions discussed. For instance, I would expect that if a factor has a profile that varies with time, one would expect that this would be under-represented in the unconstrained PMF solution (with some of its variability being represented by other factors) and therefore under-represented in the ME-2 solution.

REPLY: Both of the PMF and ME-2 methods assume that the source profiles are invariant with time during the whole campaigns, and the invariant source profiles identified by PMF or ME-2 are the relatively best selection to the final results in terms of statistics. In order to prove the improvement of using the anchor profiles generated by the unconstrained PMF run with the same local datasets, which do not depend on other studies, we also run the ME-2 analysis using the anchor profiles in the literature, with the results shown in Table 1-2. For Qingdao, the correlations between POAs and their tracers and the Q/Qexp values using the three BBOA profiles in the literature are poorer than using the BBOA obtained in this study (Table 1). For Dongguan, the results from ME-2 using the HOA profiles in the literature are also poorer than using the HOA profiles obtained

in this study (Table 2). Therefore, it can be seen that the method to get an anchor profile in this study is easier (it does not depend on the literature) and more valid. We have added the above analysis in section 3 in the revised manuscript.

3. A step-by-step recommended procedure should be unambiguously presented, for the benefit of those attempting to recreate the method. While this is kind-of done in the conclusions, it is very vague in places.

REPLY: We have adjusted the relevant text structures accordingly.

4. As a final technical query, can the authors confirm that the PAH data used to validate the result were not allowed to influence the factorisation originally? It would defeat the object of the exercise if they were.

REPLY: The process of PAH quantification is now added in Section 2.4. The input matrix for PMF/ME-2 in this study does not include PAH fitting ions. We generally use the matrix with m/z of less than 100 (or 150) as the PMF/ME-2 input data, but the PAHs ions mostly have m/z of above 150. Therefore, the PAH data do not influence the factorization.

References

Crippa, M., DeCarlo, P. F., Slowik, J. G., Mohr, C., Heringa, M.F., Chirico, R., Poulain, L., Freutel, F., Sciare, J., Cozic, J., DiMarco, C. F., Elsasser, M., José, N., Marchand, N., Abidi, E.,Wiedensohler, A., Drewnick, F., Schneider, J., Borrmann, S.,Nemitz, E., Zimmermann, R., Jaffrezo, J.-L., Prévôt, A. S. H.,and Baltensperger, U.: Wintertime aerosol chemical compositionand source apportionment of the organic fraction in themetropolitan area of Paris, Atmos. Chem. Phys., 13, 961–981,doi:10.5194/acp-13-961-2013, 2013.

Crippa, M., F. Canonaco, V. a. Lanz, M. Äijälä, J. D. Allan, S. Carbone, G. Capes, D. Ceburnis, M. Dall'Osto, D. A. Day, P. F. DeCarlo, M. Ehn, a. Eriksson, E. Freney, L. Hildebrandt Ruiz, R. Hillamo, J. L. Jimenez, H. Junninen, A. Kiendler-Scharr, A. M.

Kortelainen, M. Kulmala, A. Laaksonen, A. A. Mensah, C. Mohr, E. Nemitz, C. O'Dowd, J. Ovadnevaite, S. N. Pandis, T. Petäjä, L. Poulain, S. Saarikoski, K. Sellegri, E. Swietlicki, P. Tiitta, D. R. Worsnop, U. Baltensperger, and A. S. H. Prévôt: Organic aerosol components derived from 25 AMS data sets across Europe using a consistent ME-2 based source apportionment approach, Atmos. Chem. Phys., 14(12),6159-6176, doi: 10.5194/acp-14-6159-2014, 2014.

Elser, M., R. J. Huang, R. Wolf, J. G. Slowik, Q. Wang, F. Canonaco, G. Li, C. Bozzetti, K. R. Daellenbach, Y. Huang, R. Zhang, Z. Li, J. Cao, U. Baltensperger, I. El-Haddad, and A. S. H. Prévôt : New insights into PM2.5 chemical composition and sources in two major cities in China during extreme haze events using aerosol mass spectrometry, Atmos. Chem. Phys., 16(5), 3207-3225, doi: 10.5194/acp-16-3207-2016,2016.

He, L. Y., Y. Lin, X. F. Huang, S. Guo, L. Xue, Q. Su, M. Hu, S. J. Luan, and Y. H. Zhang : Characterization of high-resolution aerosol mass spectra of primary organic aerosol emissions from Chinese cooking and biomass burning, Atmos. Chem. Phys., 10(23), 11535-11543, doi: 10.5194/acp-10-11535-2010,2010.

Zheng J., Hu M., Gu F.T., Peng J.F., Zhang W.B., Xiao Y., Du Z.F., Qin Y.H., Deng L., Li M.R., Wu Y.S, Shuai S.J.: Characterization of High Resolution Source Profiles of Primary Organic Aerosol missions From Gasoline Vehicles. Proceedings of the CSEE., 36(16), 4466-4471,doi: 10.13334/j.0258-8013.pcsee.160358, 2016.